**Data Availability Statement:** All data are held in a public repository (Figshare) and will become publicly available upon acceptance of the

# Central and local arterial stiffness in White Europeans compared to age-, sex-, and BMI-matched South Asians

Koen M. van der Sluijs[1]☉*, Jos Thannhauser[1,2]☉, Iris M. Visser[1,3], P. M. Nabeel[4], Kiran V. Raj[5], Afrah E. F. Malik[6], Koen D. Reesink[6], Thijs M. H. Eijsvogels[1], Esmée A. Bakker[1], Prabhdeep Kaur[7], Jayaraj Joseph[5], Dick H. J. Thijssen[1]

1 Department of Medical BioSciences, Radboud University Medical Center, Nijmegen, Gelderland, The Netherlands, 2 Faculty of Science and Technology, Department of Cardiovascular and Respiratory Physiology, University of Twente, Enschede, Overijssel, The Netherlands, 3 Technical Medicine, University of Twente, Enschede, Overijssel, The Netherlands, 4 Healthcare Technology Innovation Centre, Indian Institute of Technology Madras, Chennai, Tamil Nadu, India, 5 Department of Electrical Engineering, Indian Institute of Technology Madras, Chennai, Tamil Nadu, India, 6 Department of Biomedical Engineering, CARIM School for Cardiovascular Diseases, Maastricht University Medical Center, Maastricht, Limburg, The Netherlands, 7 National Institute of Epidemiology, Indian Council of Medical Research, Chennai, Tamil Nadu, India

☉ These authors contributed equally to this work.
* Koen.vanderSluijs@radboudumc.nl

## Abstract

### Background

Ethnicity impacts cardiovascular disease (CVD) risk, and South Asians demonstrate a higher risk than White Europeans. Arterial stiffness is known to contribute to CVD, and differences in arterial stiffness between ethnicities could explain the disparity in CVD risk. We compared central and local arterial stiffness between White Europeans and South Asians and investigated which factors are associated with arterial stiffness.

### Methods

Data were collected from cohorts of White Europeans (the Netherlands) and South Asians (India). We matched cohorts on individual level using age, sex, and body mass index (BMI). Arterial stiffness was measured with ARTSENS® Plus. Central stiffness was expressed as carotid-femoral pulse wave velocity (cf-PWV, m/s), and local carotid stiffness was quantified using the carotid stiffness index (Beta) and pressure-strain elastic modulus (Epsilon, kPa). We compared arterial stiffness between cohorts and used multivariable linear regression to identify factors related to stiffness.

### Results

We included n = 121 participants per cohort (age 53±10 years, 55% male, BMI 24 kg/m$^2$). Cf-PWV was lower in White Europeans compared to South Asians (6.8±1.9 vs. 8.2±1.8 m/s, p<0.001), but no differences were found for local stiffness parameters Beta (5.4±2.4 vs. 5.8±2.3, p = 0.17) and Epsilon (72±35 vs. 70±31 kPa, p = 0.56). Age (standardized β, 95%

manuscript. DOI: https://doi.org/10.6084/m9.figshare.22332034.

**Funding:** A.E.F.M. was supported by the European Union-funded Horizon 2020 project InSiDe (Grant No. 871547). The funders had no role in study design, data collection and analysis, decision to publish, or preparation of the manuscript.

**Competing interests:** The authors have declared that no competing interests exist.

confidence interval: 0.28, 0.17–0.39), systolic blood pressure (0.32, 0.21–0.43), and South Asian ethnicity (0.46, 0.35–0.57) were associated with cf-PWV; associations were similar between cohorts (p>0.05 for interaction). Systolic blood pressure was associated with carotid stiffness in both cohorts, whereas age was associated to carotid stiffness only in South Asians and BMI only in White Europeans.

## Conclusion

Ethnicity is associated with central but not local arterial stiffness. Conversely, ethnicity seems to modify associations between CVD risk factors and local but not central arterial stiffness. This suggests that ethnicity interacts with arterial stiffness measures and the association of these measures with CVD risk factors.

## Introduction

Cardiovascular disease (CVD) is the world's leading cause of morbidity and mortality [1, 2]. The prevalence of CVD is known to vary across different ethnic groups [3, 4]. South Asians demonstrate a higher burden of conventional CVD risk factors than White populations [5–8]. Traditional CVD risk factors (e.g., obesity, type 2 diabetes mellitus, dyslipidemia, hypertension, and tobacco use) and CVD events present themselves at a younger age [7–11], and contemporary risk prediction models underestimate CVD risk in South Asians [12–14]. Possibly, other factors play a role in CVD development in South Asians, which may be reflected by a distinct impact of ethnicity on vascular health [3, 10, 15].

Central arterial stiffness, quantified by carotid-femoral pulse wave velocity (cf-PWV), is an early marker of impaired vascular health [16, 17]. Cf-PWV is known to predict cardiovascular events and all-cause mortality [17–22]. Previous work evaluated ethnicity-based differences in cf-PWV to better understand disparities in CVD risk between South Asian and White populations [3, 23–28]. These studies demonstrated inconsistent findings, with cf-PWV reported to be lower [3], not different [23–25], or higher [26–28] in South Asians compared to White populations. In addition to central arterial stiffness, studies have explored local carotid stiffness, such as carotid distensibility and elasticity. Although studies reported ethnicity-based differences in local stiffness [29–32], previous work did not include South Asians. A comprehensive evaluation of central and local stiffness, combined with matching of participants at individual level, may provide better insight into the impact of ethnicity.

To this extent, we first compared central and local arterial stiffness between White Europeans from the Netherlands and South Asians from India. Second, we identified factors associated with central and local arterial stiffness and evaluated whether these factors differ between the populations to better understand the impact of ethnicity on CVD risk.

## Materials and methods

### Study populations

Data from the White European cohort were collected as part of an ongoing Dutch prospective cohort study (Nijmegen Exercise Study). Adult volunteers were recruited in May 2021, and n = 265 individuals were included between May and September 2021. Ethical approval was obtained from the local Medical Research Ethics Committee (NL36743.091.11).

Data from the South Asian cohort were derived from an Indian cross-sectional study (n = 1,074) executed in the Tiruvallur district, Tamil Nadu, South India. Volunteers (age ≥30 years) were recruited and included between August 2017 and August 2018. Ethical approval was obtained from the Institutional Human Ethics Committee (NIE/IHEC/201407-02). Both studies were carried out in accordance with the Declaration of Helsinki, and all participants provided written informed consent. Additional information regarding the ethical, cultural, and scientific considerations specific to inclusivity in global research is included in the Supporting Information (S2 Checklist).

We performed one-to-one matching of the populations based on sex, age, and body mass index (BMI). We performed exact matching on sex and accepted a maximum age difference of 5 years and a maximum BMI difference of 2.5 kg/m$^2$.

## Data collection

General patient characteristics were derived from questionnaires, including age, sex, tobacco use, and medical history [33, 34]. As for medical history, we inquired the presence of physician-made diagnoses of diabetes mellitus, hypertension, hypercholesterolemia, and history of CVD events, including myocardial infarction, stroke, thrombosis, heart failure, and cardiopulmonary resuscitation.

Prior to the study procedures, participants were instructed to fast (White Europeans: 4 hours; South Asians: 10 hours) and to abstain from alcohol and caffeinated drinks for at least 18 hours, in line with the ARTERY society guidelines for the assessment of arterial pulse wave velocity [35]. Height (cm), weight (kg), and waist and hip circumferences (cm) were measured (Seca GmbH & Co. KG, Hamburg, Germany), from which body mass index (BMI, kg/m$^2$) and waist-to-hip ratio (WHR) were calculated. Venous serum was used to analyze total cholesterol, high-density lipoprotein (HDL) cholesterol, low-density lipoprotein (LDL) cholesterol, and glucose (all in mmol/L).

## Assessment of arterial stiffness

Arterial stiffness measurements were performed with ARTSENS® Plus (Healthcare Technology Innovation Centre, Indian Institute of Technology Madras, Chennai, India), a recently developed and validated, noninvasive, image-free ultrasound device to assess central and local arterial stiffness [36–38]. The ARTSENS® Plus uses a single-element ultrasound probe and a femoral pressure cuff for the simultaneous recording of carotid diameter and femoral pressure pulse waveforms, from which real-time arterial stiffness parameters are evaluated. The A-mode data captured by the ultrasound probe are processed with validated automated algorithms to yield carotid diameter [39, 40].

After a 5-minute resting period in supine position, patient characteristics (age, sex, height, and weight) were recorded in the ARTSENS® Plus. Dedicated blood pressure cuffs were attached around the participant's left upper arm and left thigh (widths of 12.5 cm and 14.0 cm, respectively), ensuring a tight and fixed fit. Noninvasive brachial systolic blood pressure (SBP), diastolic blood pressure (DBP), and heart rate were measured once by a medical-grade blood pressure monitor integrated in the ARTSENS® Plus. Three straight distances (mm) were measured with a tape measure to estimate the effective path length (*D*) travelled by the pulse wave, according to established methods [35, 41, 42]. The first distance (*i*) was measured between the sternal notch and the left common carotid artery site as identified by palpation. The second distance (*ii*) was measured between the sternal notch and the top of the thigh cuff. The third distance (*iii*) was measured between the top of the thigh cuff and the groin area where the femoral artery could be palpated. The three distances were recorded in the ARTSENS® Plus, which then estimated the effective path length as: *D* = *ii*−*i*−*iii*.

Next, the operator positioned the gel-covered ultrasound probe at the left common carotid artery site that was approximated earlier. Guided by the display of the A-mode ultrasound signal, the operator reoriented the probe such that strong and distinct echoes of the arterial walls were visible, allowing the software to recognize them and track the artery distension waveform. Simultaneously, the thigh cuff automatically inflated to sub-diastolic pressure to capture the femoral artery pressure waveforms. The measurement was completed after capturing ten synchronously-measured high-quality carotid artery distension cycles and femoral artery pressure waveforms. The pulse transit time was defined as the time delay between the feet of the carotid and femoral artery waveforms and was averaged over the ten cycles [38].

Measurements were performed by trained operators (White Europeans: single measurement by one of five operators; South Asians: duplicate measurement by two operators and average values used for analysis). The primary outcome measures were cf-PWV (m/s) for central arterial stiffness and carotid stiffness index (Beta) and pressure-strain elastic modulus (Epsilon, kPa) for local carotid stiffness.

## Statistics

Data were pseudonymized; only the authors involved in the data management procedures had access to information that could identify individual participants. Statistical analyses were performed with IBM SPSS Statistics for Windows, version 27 (IBM Corp., Armonk, N.Y., USA). For all analyses, $p < 0.05$ was considered statistically significant. The figures presented in this article were created with RStudio (version 4.1.3) using the packages ggplot2 and forestplot. Continuous variables were visually inspected for Gaussian distribution using histograms, reported as means ± standard deviations and compared between the cohorts using independent sample t-tests. Categorical variables were reported as numbers (%) and compared between the cohorts using Fisher's exact test.

Univariable linear regression analyses were performed with cf-PWV as a dependent variable and ethnicity, age, sex, BMI, hypertension, history of CVD event, tobacco use, SBP, DBP, total cholesterol, HDL cholesterol, LDL cholesterol, total-to-HDL cholesterol ratio, and glucose as independent variables. We only selected variables for which data were available in at least 80% of the participants of each cohort, and we only selected categorical variables with a minimum of five observations per category. Second, we used forward-stepwise entry to create a multivariable linear regression model of main effects. Only variables with $p < 0.1$ in the univariable regression analyses were offered for stepwise entry. Visual inspection of the residuals was performed to check linearity. Multicollinearity of the variables included in the model was assessed using the variance inflation factor. We expanded the model by entering interaction terms between ethnicity and the independent variables included in the model to evaluate ethnicity-based differences. In case of a significant interaction term, we performed stratified analyses per cohort. Similar procedures were performed for the local arterial stiffness parameters, i.e. Beta and Epsilon.

## Results

A total of n = 242 participants were included, with n = 121 participants in both the White European and South Asian cohorts. Age was 53±10 years, and 67 (55%) participants were male in both groups. BMI was 24.1±3.3 and 23.9±3.7 kg/m² in the White European and South Asian cohorts, respectively. Compared to White Europeans, South Asians were more often current smokers, had a higher WHR, and had higher glucose levels (Table 1). South Asians had lower SBP and DBP values and lower total and HDL cholesterol levels compared to White

**Table 1. Participant characteristics of the study cohorts.**

|  | White Europeans | South Asians | p-value |
|---|---|---|---|
|  | n = 121 | n = 121 |  |
| **Matched variables** |  |  |  |
| Age, yrs | 53±10 | 53±10 | 0.93 |
| Male sex | 67 (55) | 67 (55) | >0.99 |
| Body mass index, kg/m$^2$ | 24.1±3.3 | 23.9±3.7 | 0.66 |
| **Cardiovascular risk factors** |  |  |  |
| Diabetes mellitus * | 1 (1) | 5 (4) | 0.21 |
| Hypertension * | 11 (9) | 11 (9) | >0.99 |
| Hypercholesterolemia ¤ | 9 (8) | 3 (2) | 0.08 |
| History of CVD event *,‡ | 8 (7) | N/A | - |
| Tobacco use + |  |  | <0.001 |
| Current user | 7 (6) | 14 (12) |  |
| Former user | 33 (28) | 6 (5) |  |
| Never | 79 (66) | 101 (83) |  |
| **Measurement characteristics** |  |  |  |
| Height, cm | 176±9 | 161±9 | <0.001 |
| Weight, kg | 75±14 | 62±12 | <0.001 |
| Waist-to-hip ratio § | 0.87±0.08 | 0.97±0.04 | <0.001 |
| Systolic blood pressure, mmHg | 132±15 | 121±21 | <0.001 |
| Diastolic blood pressure, mmHg | 79±9 | 75±13 | 0.010 |
| Total cholesterol, mmol/l ¤ | 5.2±0.9 | 4.8±1.1 | 0.002 |
| HDL cholesterol, mmol/l ¤ | 1.6±0.4 | 1.2±0.3 | <0.001 |
| LDL cholesterol, mmol/l ¤ | 3.1±0.8 | 3.3±0.9 | 0.06 |
| Total cholesterol/HDL ratio ¤ | 3.3±0.8 | 4.3±1.2 | <0.001 |
| Glucose, mmol/l ¤ | 4.9±0.5 | 5.8±1.3 | <0.001 |

Abbreviations: CVD: cardiovascular disease, HDL: high-density lipoprotein, LDL: low-density lipoprotein, N/A: not available. Continuous variables are presented as means ± standard deviations; categorical variables are expressed as n (%).

* n = 118 for White European cohort.

¤ n = 117 for White European cohort.

‡ Includes myocardial infarction, stroke, thrombosis, heart failure, and cardiopulmonary resuscitation.

+ n = 119 for White European cohort.

§ n = 46 for White European cohort; n = 120 for South Asian cohort.

Europeans. LDL cholesterol level and prevalence of diabetes or hypercholesterolemia did not differ between cohorts (Table 1).

## Comparison of arterial stiffness between cohorts

Cf-PWV was significantly lower in the White European cohort compared to the South Asian cohort (6.8±1.9 vs. 8.2±1.8 m/s, p<0.001). No differences in local arterial stiffness were found between White Europeans and South Asians (Beta: 5.4±2.4 vs. 5.8±2.3, p = 0.17; Epsilon: 72 ±35 vs. 70±31 kPa, p = 0.56, respectively) (Fig 1).

## Factors associated with arterial stiffness

Univariable linear regression revealed significant associations of cf-PWV, Beta, and Epsilon with the independent variables (S1–S3 Tables). In all subsequent multivariable linear regression models, linearity was assumed based on visual inspection of residuals.

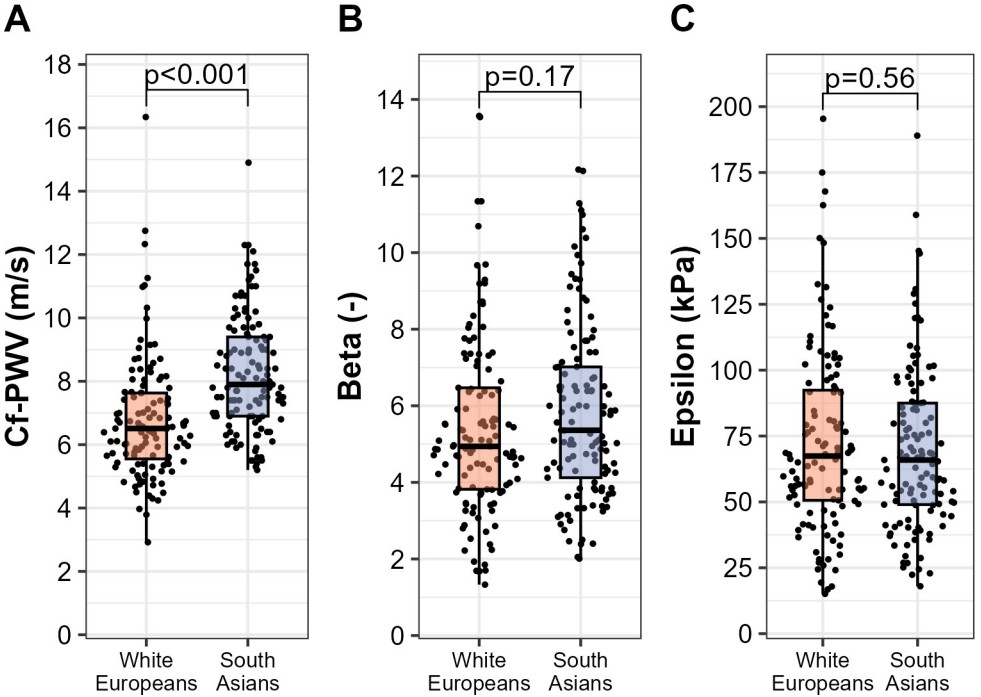

**Fig 1. Central and local arterial stiffness for the White European (n = 121) and South Asian (n = 121) cohorts.** A) Carotid-femoral pulse wave velocity (cf-PWV, m/s). B) Carotid stiffness index Beta (dimensionless). C) Pressure-strain elastic modulus Epsilon (kPa). Each dot represents an individual data point whereas box plots represent group statistics (Q1, median, Q3; whiskers extending up to 1.5 times the interquartile range). Differences between groups were assessed with an independent sample t-test.

**Cf-PWV.** The multivariable model without interaction terms with cf-PWV as dependent variable demonstrated associations with age (standardized coefficient β = 0.28; 95% confidence interval (CI): [0.17, 0.39]; p<0.001), SBP (β = 0.32; CI: [0.21, 0.43]; p<0.001), and South Asian ethnicity (β = 0.46; CI: [0.35, 0.57]; p<0.001) (Fig 2). The associations were not different between ethnicities (p>0.05 for interactions).

**Beta.** The multivariable model for carotid stiffness index Beta showed associations with age (β = 0.18; CI: [0.06, 0.31]; p = 0.005), SBP (β = 0.18; CI: [0.05, 0.31]; p = 0.007), and BMI (β

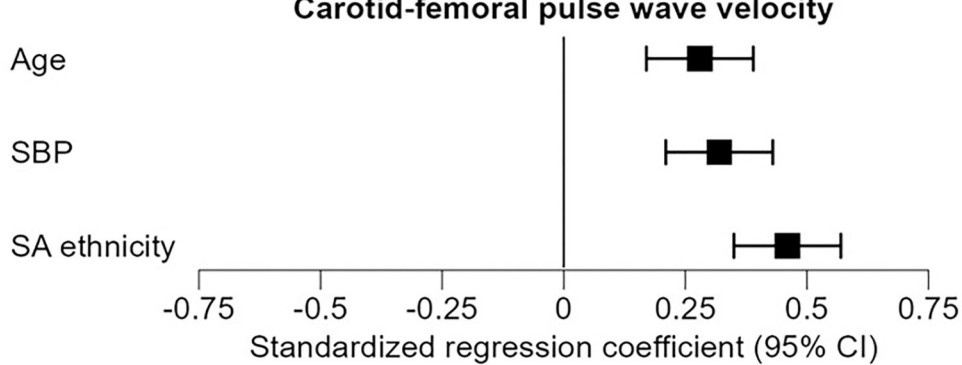

**Fig 2. Associations with central arterial stiffness parameter carotid-femoral pulse wave velocity.** Estimates of the standardized regression coefficients of the multivariable model are shown; error bars represent the 95% confidence intervals. Abbreviations: CI: confidence interval, SA: South Asian, SBP: systolic blood pressure.

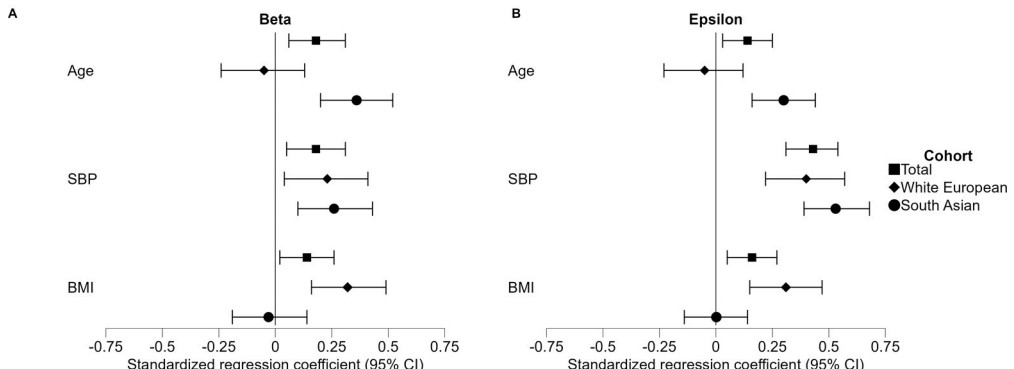

**Fig 3.** Associations with local arterial stiffness parameters A) carotid stiffness index Beta, and B) pressure-strain elastic modulus Epsilon. Estimates of the standardized regression coefficients of the multivariable model are shown; error bars represent the 95% confidence intervals. Abbreviations: BMI: body mass index, CI: confidence interval, SBP: systolic blood pressure.

= 0.14; CI: [0.02, 0.26]; p = 0.026). Significant interaction terms were found between ethnicity and age and BMI, so separate models were created for the White European cohort and South Asian cohort. The stratified analysis revealed that BMI was associated with carotid stiffness index Beta in the White European cohort but not in the South Asian cohort. Moreover, age was significantly associated to carotid stiffness index Beta in the South Asian cohort but not in the White European cohort (Fig 3A).

**Epsilon.**   In the multivariable model for pressure-strain elastic modulus Epsilon, we found associations with age (β = 0.14; CI: [0.03, 0.25]; p = 0.017), SBP (β = 0.43; CI: [0.31, 0.54]; p<0.001), and BMI (β = 0.16; CI: [0.05, 0.27]; p = 0.006). Because of significant interaction terms between ethnicity and age and BMI, we created cohort-specific models. The stratified analysis showed a significant association of Epsilon with BMI in the White European cohort and revealed that Epsilon was associated to age in the South Asian cohort (Fig 3B).

## Discussion

We compared central and local arterial stiffness between a White European and a matched South Asian cohort. We found a significantly lower cf-PWV in the White European cohort compared to the South Asian cohort, whilst no such differences were found for local arterial stiffness. Second, we identified factors associated with central and local arterial stiffness (e.g., age, SBP, and BMI), and we investigated whether these associations differed between ethnic cohorts. Central arterial stiffness was significantly associated with age and SBP, and we found that ethnicity did not alter this relation. In contrast, we found ethnicity-dependent associations for local arterial stiffness. Age was associated with local stiffness only in South Asians, and BMI was associated with local stiffness only in White Europeans, whilst SBP was associated with local stiffness in both cohorts. Altogether, our findings suggest that ethnicity is associated with central, but not local arterial stiffness. Furthermore, ethnicity seems to alter the impact of factors associated with local, but not central arterial stiffness. This supports the existing evidence that ethnicity interacts with measures of arterial stiffness and adds new insight into the influence of ethnicity on factors related to CVD risk.

Our finding of a higher cf-PWV in South Asians is consistent with some [26–28], but in contrast with other studies [3]. In line with our findings, two small studies based on United Kingdom cohorts found higher cf-PWV values in South Asians than in White Europeans [27, 28]. Similarly, a Dutch study reported a higher cf-PWV in South Asians than in White

Europeans aged >40 years [26]. In contrast, a large American study found a lower cf-PWV in South Asians than in the resident White population [3]. Moreover, some studies found no differences between the two populations [23–25]. These aforementioned studies are methodologically similar to our present study pertaining to sample size (range 131–4,211 participants), use of validated methods for measuring cf-PWV, and adjustment for conventional CVD risk factors. A possible explanation for the discrepancy in findings is that cf-PWV may be influenced by ethnic-based differences that were not adjusted for, such as WHR, body surface area, or proinflammatory state [7]. One important difference between studies is that we included participants residing in the countries of their ethnic origin. Whilst this may have introduced a between-country bias, it allowed us to evaluate the impact of ethnicity on arterial stiffness in the original ethnic settings.

By investigating which factors are related to central arterial stiffness, we found that only higher age and SBP were positively associated with cf-PWV. The relations between cf-PWV and age and SBP have been described previously [17, 22, 26, 43]. For example, a systematic review of 77 studies reported that age and SBP are strongly related to cf-PWV [43]. Notably, this review also reported that other CVD risk factors are not or only modestly related to cf-PWV, an observation that matches our findings. We add the insight that the association between age and central stiffness was not ethnicity-dependent, which agrees with the American study [3] but disagrees with the Dutch study [26]. In the latter study, the positive relation between age and cf-PWV was significantly greater for the South Asian group than for the Dutch group, resulting in a greater cf-PWV difference between ethnicities at higher age. Furthermore, we found that cf-PWV was significantly associated with age, an effect that was similarly present in both ethnic groups. This suggests that ethnicity does not alter the impact of risk factors on measures of central arterial stiffness. Altogether, our findings corroborate previous work and suggest that factors related to central arterial stiffness are robust and do not differ in direction and/or magnitude between South Asian and White European ethnicities.

To our knowledge, this study is the first to compare local carotid stiffness between these two populations. In contrast to our findings on central arterial stiffness, we found no differences between the White European and South Asian cohorts in measures of local arterial stiffness, which is somewhat unexpected. However, it fits with the evidence that central and local arterial stiffness are distinct measures that cannot be used interchangeably [44]. As both central and local measures are known to predict incident cardiovascular events and all-cause mortality [19, 44], combining central and local arterial stiffness measures may provide the most predictive information on vascular health status.

Although local carotid stiffness did not differ between groups, we found age, SBP, and BMI to relate to local carotid stiffness. The relation between higher carotid stiffness and older age [45–47] as well as the association between carotid stiffness and BMI have been described before [48–50]. Interestingly, we found that ethnicity impacted the associations for both local stiffness parameters. Specifically, we found that BMI was related to carotid stiffness in White Europeans, whilst age was associated with carotid stiffness in South Asians. Possibly, the distinct associations of BMI with local stiffness between cohorts may relate to differences in body composition. South Asians tend to have more visceral adipose tissue and a higher fat percentage at a given BMI level compared to White Europeans [7, 51, 52], which seems to alter the relation between BMI and CVD risk [53, 54]. Differences in body composition between cohorts may therefore contribute to the differences we observed in the association of BMI with local stiffness. The difference between cohorts in the association of age with carotid stiffness may, at least in part, be related to the earlier onset of CVD in South Asians compared to White Europeans [7, 8, 11]. As our study includes relatively young participants, the relation between age and carotid stiffness may have developed in the South Asian cohort but not yet in the

White European cohort. Taken together, our observations suggest that ethnicity does not alter local carotid stiffness per se, but it may affect the underlying factors related to local stiffness.

Strengths of our study are the use of the same validated device to assess central and local arterial stiffness in both cohorts [37, 38], the inclusion of a large group of participants, and the inclusion of participants residing in the country of their ethnic origin. However, some limitations are present. A potential limitation is a selection bias introduced by the matching procedure. From the 265 and 1,074 individuals in the Dutch and Indian cohorts, respectively, 121 participants were included per cohort after matching. However, participant characteristics indicated that these samples represent the source populations well, so we expect the potential selection bias to be minimal. Another limitation is that data on CVD history in the South Asian cohort were not collected. This prevented us from providing a complete overview of CVD risk profile of both cohorts and from investigating whether this introduced a bias in our results. However, we incorporated conventional cardiovascular risk factors (e.g., the presence of hypertension or hypercholesterolemia, smoking behavior, and blood pressure) in our analyses and found no important differences. Therefore, we expect that the lack of this information does not impact our main outcomes.

In conclusion, we found that central arterial stiffness was lower in White Europeans compared to South Asians, but ethnicity did not affect the associations of risk factors (e.g., age, systolic blood pressure) and central stiffness. In contrast, we found no direct impact of ethnicity on local measures of arterial stiffness, whereas ethnicity did alter the associations between risk factors and local arterial stiffness. Therefore, our work demonstrates that ethnicity is related to measures of arterial stiffness and to the associations between these measures and CVD risk factors. Further research is warranted to investigate whether these ethnicity-based differences in central and local arterial stiffness have consequences for the development of CVD-related outcomes.

## Supporting information

**S1 Checklist. STROBE statement.**
(DOCX)

**S2 Checklist. Inclusivity in global research questionnaire.**
(DOCX)

**S1 Table. Univariable regression coefficients for the associations with carotid-femoral pulse wave velocity.**
(DOCX)

**S2 Table. Univariable regression coefficients for the associations with carotid stiffness index Beta.**
(DOCX)

**S3 Table. Univariable regression coefficients for the associations with pressure-strain elastic modulus Epsilon.**
(DOCX)

**S4 Table. Pulse transit time and distances for calculating carotid-femoral pulse wave velocity, per cohort.**
(DOCX)

## Acknowledgments

The authors sincerely thank all personnel involved in the data collection.

## Author Contributions

**Conceptualization:** Koen M. van der Sluijs, Jos Thannhauser, Jayaraj Joseph, Dick H. J. Thijssen.

**Data curation:** Koen M. van der Sluijs, Jos Thannhauser, Iris M. Visser, P. M. Nabeel, Kiran V. Raj.

**Formal analysis:** Koen M. van der Sluijs, Jos Thannhauser, Iris M. Visser.

**Funding acquisition:** Jayaraj Joseph, Dick H. J. Thijssen.

**Investigation:** Koen M. van der Sluijs, Jos Thannhauser, P. M. Nabeel, Kiran V. Raj, Prabhdeep Kaur.

**Methodology:** Koen M. van der Sluijs, Jos Thannhauser, Jayaraj Joseph, Dick H. J. Thijssen.

**Project administration:** Koen M. van der Sluijs, Jos Thannhauser, P. M. Nabeel, Kiran V. Raj.

**Resources:** Koen D. Reesink, Thijs M. H. Eijsvogels, Prabhdeep Kaur, Jayaraj Joseph, Dick H. J. Thijssen.

**Software:** P. M. Nabeel, Kiran V. Raj, Jayaraj Joseph.

**Supervision:** Thijs M. H. Eijsvogels, Esmée A. Bakker, Jayaraj Joseph, Dick H. J. Thijssen.

**Validation:** Koen M. van der Sluijs, Jos Thannhauser, Iris M. Visser.

**Visualization:** Koen M. van der Sluijs, Jos Thannhauser, Iris M. Visser.

**Writing – original draft:** Koen M. van der Sluijs, Jos Thannhauser, Iris M. Visser, Dick H. J. Thijssen.

**Writing – review & editing:** P. M. Nabeel, Kiran V. Raj, Afrah E. F. Malik, Koen D. Reesink, Thijs M. H. Eijsvogels, Esmée A. Bakker, Prabhdeep Kaur, Jayaraj Joseph.

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
