## [Decision Letter · Decision Letter 0]

19 Jun 2023

PONE-D-23-08879Central and local arterial stiffness in White Europeans compared to age-, sex-, and BMI-matched South AsiansPLOS ONE

Dear Dr. van der Sluijs,

Thank you for submitting your manuscript to PLOS ONE. After careful consideration, we feel that it has merit but does not fully meet PLOS ONE’s publication criteria as it currently stands. Therefore, we invite you to submit a revised version of the manuscript that addresses the points raised during the review process.

We look forward to receiving your revised manuscript.

Kind regards,

Giacomo Pucci

Academic Editor

PLOS ONE

Journal Requirements:

Reviewers' comments:

Reviewer's Responses to Questions

**Comments to the Author**

1. Is the manuscript technically sound, and do the data support the conclusions?

Reviewer #1: Partly

2. Has the statistical analysis been performed appropriately and rigorously? 

Reviewer #1: Yes

3. Have the authors made all data underlying the findings in their manuscript fully available?

Reviewer #1: Yes

4. Is the manuscript presented in an intelligible fashion and written in standard English?

Reviewer #1: Yes

5. Review Comments to the Author

Reviewer #1: The authors analysed data of central (aortic) and local (carotid) arterial stiffness in two matched cohorts of Europeans and South Asians. They found that ethnicity is associated with central but not local arterial stiffness.

This study provides interesting insights into ethnic-based differences in cardiovascular phenotypes.

My remarks are as follows:

Introduction: In the first parpagraph, the authors refer to atherosclerosis, while arterial stiffness rather refers to arteriosclerosis, a pathological process different but related to atherosclerosis. The correct terminology and also definitions of these two conditions should be used in the manuscript.

Methods: I wonder why measurements from Europeans and South Asians were treated differently, as European had only one measurement and South Asians the mean of two. This could have reduced the variability of data among South Asians. If possible, the same anlytical method should be used for both cohorts.

Methods: It is not very clear how the ARTSENS device uses the three distances measured over the body surface to calculate cfPWV. Please clarify this in methods as readers may not be familiar with this device. Please add mean transit times and distances for each group in the supplementary material.

Methods: Are there any validation data for ARTSENS device in literature?

Results and discussion: Potential confoundinf factors for cfPWV differences between ethnicities may have been anthropometric differences (height in particular) and the prevalence of smoking. These two variables should be added to univariate and multivariate analysis. Based on the results of this added analysis, the importance of these two factors should be discussed.

6. PLOS authors have the option to publish the peer review history of their article (what does this mean?). If published, this will include your full peer review and any attached files.

Reviewer #1: No

---

## [Author Response · Author response to Decision Letter 0]

12 Jul 2023

Dear Prof. Dr. Pucci,

Thank you for your email on June 19th concerning our manuscript entitled “Central and local arterial stiffness in White Europeans compared to age-, sex-, and BMI-matched South Asians”. We greatly appreciate the opportunity to revise our manuscript and have enclosed a revised version. In addition, a file marked with track changes is appended. 

We believe we have adequately addressed the comments. A point-by-point response to the Editor’s and reviewer’s comments is enclosed below. 

Sincerely yours, on behalf of the authors,

Koen van der Sluijs, MSc

Corresponding Author

---

## [Decision Letter · Decision Letter 1]

2 Aug 2023

Central and local arterial stiffness in White Europeans compared to age-, sex-, and BMI-matched South Asians

PONE-D-23-08879R1

Dear Dr. van der Sluijs,

We’re pleased to inform you that your manuscript has been judged scientifically suitable for publication and will be formally accepted for publication once it meets all outstanding technical requirements.

Kind regards,

Giacomo Pucci

Academic Editor

PLOS ONE

Additional Editor Comments (optional):

Reviewers' comments:

Reviewer's Responses to Questions

**Comments to the Author**

1. If the authors have adequately addressed your comments raised in a previous round of review and you feel that this manuscript is now acceptable for publication, you may indicate that here to bypass the “Comments to the Author” section, enter your conflict of interest statement in the “Confidential to Editor” section, and submit your "Accept" recommendation.

Reviewer #1: All comments have been addressed

2. Is the manuscript technically sound, and do the data support the conclusions?

Reviewer #1: Yes

3. Has the statistical analysis been performed appropriately and rigorously? 

Reviewer #1: Yes

4. Have the authors made all data underlying the findings in their manuscript fully available?

Reviewer #1: Yes

5. Is the manuscript presented in an intelligible fashion and written in standard English?

Reviewer #1: Yes

6. Review Comments to the Author

Reviewer #1: The authors adequately answered to my previous remarks.

The manuscript seems now suitable for publication.

7. PLOS authors have the option to publish the peer review history of their article (what does this mean?). If published, this will include your full peer review and any attached files.

Reviewer #1: **Yes: **Andrea Grillo

---

## [Editor Report · Acceptance letter]

15 Aug 2023

PONE-D-23-08879R1 

Central and local arterial stiffness in White Europeans compared to age-, sex-, and BMI-matched South Asians 

Dear Dr. van der Sluijs:

I'm pleased to inform you that your manuscript has been deemed suitable for publication in PLOS ONE. Congratulations! Your manuscript is now with our production department. 

Kind regards, 

on behalf of

Dr. Giacomo Pucci 

Academic Editor

PLOS ONE